# Cytochrome P450 Can Epoxidize an Oxepin to a Reactive 2,3-Epoxyoxepin Intermediate: Potential Insights into Metabolic Ring-Opening of Benzene

**DOI:** 10.3390/molecules25194542

**Published:** 2020-10-03

**Authors:** Holly M. Weaver-Guevara, Ryan W. Fitzgerald, Noah A. Cote, Arthur Greenberg

**Affiliations:** 1Department of Chemistry, Grove City College, Grove City, PA 16127, USA; 2Department of Chemistry, University of New Hampshire, Durham, NH 03824, USA; rwj9@wildcats.unh.edu (R.W.F.); nac1009@wildcats.unh.edu (N.A.C.)

**Keywords:** 2,3-epoxyoxepins, metabolic ring opening of benzene, cytochrome P450 isozymes

## Abstract

Dimethyldioxirane epoxidizes 4,5-benzoxepin to form the reactive 2,3-epoxyoxepin intermediate followed by very rapid ring-opening to an *o*-xylylene that immediately isomerizes to the stable product 1*H*-2-benzopyran-1-carboxaldehyde. The present study demonstrates that separate incubations of 4,5-benzoxepin with three cytochrome P450 isoforms (2E1, 1A2, and 3A4) as well as pooled human liver microsomes (pHLM) also produce 1*H*-2-benzopyran-1-carboxaldehyde as the major product, likely via the 2,3-epoxyoxepin. The reaction of 4,5-benzoxepin with cerium (IV) ammonium nitrate (CAN) yields a dimeric oxidized molecule that is also a lesser product of the P450 oxidation of 4,5-benzoxepin. The observation that P450 enzymes epoxidize 4,5-benzoxepin suggests that the 2,3-epoxidation of oxepin is a major pathway for the ring-opening metabolism of benzene to muconaldehyde.

## 1. Introduction

Benzene (**1**) is a very widespread environmental contaminant and human carcinogen with a complex metabolic fate (Scheme 1). The discovery [1] of the equilibrium between benzene oxide (**2**) and oxepin (**6**) laid the groundwork for the study by Davies and Whitham [2] in which they oxidized oxepins by employing meta-chloroperbenzoic acid (m-CPBA), a study relevant to the metabolism of benzene (**1**). The oxidation reactions produced acyclic dicarbonyl compounds (e.g., **5** through putative 2,3-epoxyoxepins (**7**, see Scheme 1). However, the reactive 2,3-epoxyoxepin intermediates could neither be isolated nor observed spectroscopically [2], due, in part, to the acidic conditions of their generation. The present study makes a comparison between the oxidative metabolism of benzene through benzene oxide/oxepin with that of a “non-natural metabolite” of naphthalene: 4,5-benzoxepin. It is worthwhile to very briefly compare these metabolic pathways.

Benzene is a very widespread environmental contaminant and human carcinogen with a complex metabolic fate (Scheme 1) [3,4,5,6]. It is metabolized in the liver, primarily by cytochrome P450 2E1, mostly through epoxidation to benzene oxide (**2**) [7,8,9,10], followed by rearrangement to phenol (**3**) and subsequent further oxidation as well as bioconjugation reactions. A relatively small fraction (up to 10–20%) is metabolically ring opened by P450 to *Z*,*Z*-muconaldehyde (*Z*,*Z*-**5**) which isomerizes to *E*,*E*-**5**, which is oxidized by aldehyde dehydrogenase to *E*,*E*-muconic acid (*E*,*E*-**4**, detected in urine [11]) [12,13,14,15]. Muconaldehyde is a very potent inhibitor of the normal generation of blood cells [6]. It reacts with glutathione and can cross-link DNA and proteins and induces inhibition of gap junction intracellular communication [16,17,18].

Benzene is unique in its conversion to significant quantities of closed-ring (e.g., phenol, **3**) and ring-opened (e.g., muconic acid, **4**) metabolites via mammalian monooxygenases. Alkylated aromatics such as toluene, ethylbenzene, and xylenes are predominantly metabolically oxidized at the alkyl groups (e.g., toluene to benzyl alcohol [19]). Naphthalene is metabolized to 1-naphthol. Other, larger polycyclic aromatic hydrocarbons (PAH) also form ring-substituted rather than open-chain metabolites. While there is near-universal acceptance of the benzene oxide pathway to phenol, the pathway to muconaldehyde is less understood. Direct attack by hydroxyl (OH) on benzene [20] followed by a reaction of the initial radical with molecular oxygen appears to be relevant to atmospheric formation of muconaldehyde [21]. Of particular interest is the pathway suggested by Davies and Whitham [2] nearly four decades ago. They postulated the epoxidation of oxepin to “2,3-epoxyoxepin” (**7**) followed by rapid ring opening to *Z*,*Z*-muconaldehyde (*Z*,*Z*-**5**). *Z*,*Z*-**5** isomerizes thermally as well as catalytically to *E*,*Z*-**5** followed by catalyzed isomerization to *E*,*E*-**5** (see Scheme 1) [22,23]. Using the acidic epoxidation reagents (e.g., m-CPBA) of the period, Davies and Whitham implicated the intermediacies of 2,3-epoxyoxepins without isolating or spectroscopically observing them [2].

Ab initio calculations predicted a low barrier for the concerted ring opening of **7** to *Z*,*Z*-**5** [24]. Interestingly, 3,6-bridged 2,3-epoxyoxepins were reported in 1982, synthesized using m-CPBA and exhibited melting points ca 100 °C [25]. These surprising stabilities are attributable to the influences of the bridges which both frustrate thermal rearrangements and somewhat stabilize the epoxides to acid [26]. The first report of a simple 2,3-epoxyoxepin involved reaction of 2,7-dimethyloxepin (**8**) with dimethyldioxirane (DMDO) [also methyl(trifluoromethyl)dioxirane] at −60 °C involved NMR observation of a trace amount of 2,3-epoxyoxepin **9** which isomerized to the diketone (*Z*,*Z*-**10**) at −20 °C (Scheme 2A) [27]. A similar study of the parent **2/6** equilibrium produced *Z*,*Z*-**10** but no direct observation of **7** [28]. The reaction of DMDO with 4,5-benzoxepin (**11**) at −50 °C provided a near quantitative yield of **12** which rearranged rapidly at −15 °C to form **14** almost quantitatively, undoubtedly via the short-lived ortho-xylylene **13** (Scheme 2B) [29]. 

The growing acceptance that oxepin is an intermediate in the pathway to muconaldehyde has raised other subtleties. Low-temperature DMDO reactions with oxepin, 2-methyloxepin, and 2,7-dimethyloxepin form the *Z*,*Z*-isomers (e.g., *Z*,*Z*-**5**, Scheme 3) [28]. Viewing consecutive one-electron oxidations, as another reasonable pathway to muconaldehyde, the Golding group explored surrogate reactions [30]. The reaction of the above three oxepins with cerium(IV) ammonium nitrate (CAN) produced *E*,*Z*-isomers exclusively (e.g., *E*,*Z*-**5** as well as *E*,*Z*-**10**, see Scheme 3), consistent with geometric isomerization of the ring-opened radical intermediate [30]. Without intercession of free radical intermediates, *Z*,*Z*-**5** is converted to *E*,*Z*-**5** at 55 °C in the dark in 16 h [22]. The cyclic 2*H*-pyran-2-carboxaldehyde is the higher-energy intermediate connecting these two stereoisomers. This thermal pathway does not isomerize *E*,*Z*-**5** to *E*,*E*-**5**. *E*,*Z*-**5** (as well as *Z*,*Z*-**5**) are readily converted to *E*,*E*-**5** in the presence of a nucleophilic reagent (e.g., trimethylamine in acetonitrile) acting as a catalyst in reversible Michael additions [22]. In contrast, for the 4,5-benzo-substituted system (Scheme 2), 1*H*-2-benzopyran-1-carboxaldehyde (**14**), the cyclic molecule is more stable than the *Z*,*Z*-isomer (**13**) and effectively shuts down the pathway to the *E*,*Z*-isomer, and is also higher in energy.

2,7-Dimethyloxepin (**8**, Scheme 2) is an interesting substrate for investigation of P450 metabolism. Although it is the only detectable tautomer, due to steric repulsion between the methyl substituents in the benzene oxide tautomer, the Curtin–Hammett Principle does not eliminate the undetectable tautomer from being the reactive substrate [31]. In principle, it could be derived by P450-catalyzed epoxidation of ortho-xylene. Since xylenes are oxidized at the methyl substituent, **8** should not form in significant quantity via mammalian P450 monooxygenases. One might think of it as a “non-natural metabolite further down a hypothetical pathway avoiding methyl oxidation”. 4,5-Benzoxepin (**11**) is even more interesting as a non-natural metabolite. Naturally occurring P450-catalyzed epoxidation at the 1,2-position of naphthalene (**15**) forms the 1,2-epoxide (**16**) which isomerizes to 1-naphthol (**17**) rather than to the less stable oxepin **18**, which lacks aromatic stabilization [32,33]. Hypothetical epoxidation at the 2,3-position of naphthalene would eliminate aromaticity—the 2,3-epoxide (**19**) is far less stable than the 1,2-epoxide. It is never formed, and therefore its valence isomer 4,5-benzoxepin (**11**) is not produced naturally (see Scheme 4) [32,33]. Thus, **11** can also be considered as a “non-natural metabolite further down a hypothetical pathway avoiding natural 1,2-epoxidation of naphthalene”.

Although the benzene oxide isomer of 8 is undetectable by NMR, it is only a few kcal/mol higher in energy, so that rearrangement to dimethylphenol and attack by nucleophiles should be possible. In contrast, **19** is calculated to be over 30 kcal/mol less stable than **11** [33], eliminating reactions via the epoxide. 

## 2. Results and Discussion

### 2.1. Oxidations of 4,5-Benzoxepin with DMDO and CAN

The present study compares the reaction products of DMDO and CAN oxidations of 2,7-dimethyloxepin (**8**) and 4,5-benzoxepin (**11**) with those derived from incubation with cytochrome P450 enzymes (synthetic procedures found in Appendix B and Appendix A). Whereas oxidation of 2,7-dimethyloxepin with DMDO produces *Z*,*Z*-**10** and CAN oxidation yields *E*,*Z*-**10** [29], the corresponding oxidations of 4,5-benzoxepin produced very different, non-isomeric products. A reaction with DMDO (1:1 molar ratio) produces **14** but reaction with CAN (2:1 molar ratio) produces two essentially dimeric molecules assigned structures **21** (C_20_H_16_O_4_) and **20** (C_20_H_14_O_4_) (Scheme 5). The two compounds appear to form via different pathways. As will be supported later in this paper, incubation of **11** with P450 isoforms yielded **20** as a lesser product with a minute quantity of **21**.

Spectroscopic data in support of the identifications of **20** and **21** are in the Experimental Section and further documented in the Appendix A. B3LYP/6-31G* calculations [34,35] predict that (*R*,*R*)- (or *S*,*S*)-**20** is 1.4 kcal/mol lower in energy than meso-**20**. The anti- (C_i_) diastereomer of **21** is calculated to be 5.7 kcal/mol lower in energy than the syn- (C_2v_) diastereomer. ^13^C NMR shows ten unique carbon atoms with shifts consistent with those expected for **20**. ^1^H NMR shows seven distinct protons (4H: aromatic; 2H vinylic; 1H at 6.6 ppm roughly consistent with the downfield shift calculated for the dimer (meso or *R*,*R*/*S*,*S*)). Two-dimensional NMR (COSY, DEPT, NOESY, and HSQC) suggests the presence of a conjugated alkene (NOESY correlation) and confirms the presence of one proton (6.6 ppm) on the carbon alpha to the carbonyl (DEPT, HSQC correlations). IR spectroscopy indicates a strong absorbance at 1666 cm^−1^ and no absorbance due to hydroxyl groups. The UV absorbances (λ_max_ = 273 nm and 338 nm) are also consistent with similar structural features. GC/EI-MS data as well as ESI/MS data were not definitive for a structure corresponding to **21** (expected *m*/*z* = 318, or 319 via ESI/MS). However, EI and ESI/MS are both consistent with isocoumarin which could arise from **20** through a loss of acetylene (or its equivalent, see Scheme 6A), presumably in a stepwise mechanism. Column chromatography did not provide samples of **20** or **21** suitable for elemental analysis or X-ray crystallographic study.

The EI mass spectrum observed corresponding to isocoumarin (**22**) includes an abundant *m*/*z* 90 (Parent - 2 CO) in agreement with a published mass spectrum [36]. Another published mass spectrum for isocoumarin [37] appears to erroneously report an abundant *m*/*z* 97 with no *m*/*z* 90 reported. Precise mass analysis corresponds to the formula C_9_H_6_O_2_ for the *m*/*z* 146 ion as expected for isocoumarin (see Appendix A). The ESI/MS data also correspond to isocoumarin. Further investigation of this system involved synthesis of the reported monomeric bromo derivative **23** [38]. Coupling using NiCl_2_ and Zn in the presence of pyridine and 2,2′-bipyridine to produce dimer **24** was attempted. It provided lactone **25** (Scheme 6B) in addition to reduced monomer **26**, the latter likely due to residual moisture. Although the expulsion of acetylene appears to be unlikely, it is possible that the two carbons lost en route to **25** might arise in other forms under these reaction conditions. A single small-scale headspace analysis found no acetylene and only methane as the single gaseous component, which was very surprising and is worthy of future systematic investigation.

Scheme 7 describes a postulated reaction mechanism for the formation of the dimeric **20** from 4,5-benzoxepin under CAN oxidation. While the mechanism resembles that postulated by Golding et al. [30] in its initial steps, the ring opening of the postulated free radical **25** is unfavorable due to the need to lose aromaticity; hence, **14** (via **13**) is not formed and isomerization to *E*,*Z*-**13** should not occur. While the mechanism offered in Scheme 7 depicts dimerization through the reaction of the radical **24** with **11**, one would expect analogous dimerization through **25** to be more sterically hindered. The central C–C linkage in dimeric molecule **20** might be anticipated to be somewhat weak due to the possibility of forming two capto-dative free radicals. However, a dimeric molecule capable of forming two PhCOC(OPh)(Ph) radicals showed no evidence of decomposition at 60 °C [39]. Indeed, a crystallographic study of bridged dimer containing this linkage provided a C–C bond length of 1.523 Ǻ [40]. Other stable examples are known [41,42]. The pathway to formation of **21** is suggested by the oxidation of the bridged oxepin to form **30**, presumed to occur via glycol **31** [43]. The equivalent **32** and **33** are reasonable precursors to **21** (Scheme 8). 

### 2.2. Incubations of 4,5-Benzoxepin with P450 Isoforms

Incubations of **11** with P450 2E1, P450 1A2, P450 3A4, and pooled human liver microsomes (pHLM) were compared with the products of DMDO and CAN oxidations. The 2E1 isoform is known to be optimal for benzene; the 1A2 isoform is optimal for naphthalene; the 3A4 isoform has a larger active site and is relatively non-specific. The enzymes and pHLM were obtained commercially and the co-factor mixture is described in the Experimental Section. The primary analytical method for the enzyme incubation products was GC/MS since the scale is small. 

Figure 1A reproduces the GC/MS chromatogram derived from the incubation of 4,5-benzoxepin (**11**) with cytochrome P450 1A2. Highlighted in this chromatogram are unreacted **11**, 1H-2-benzopyran-1-carboxaldehyde (**14**) and a smaller quantity of the dimer (**20**). Figure 1B reproduces the mass spectra of these three components. We have established that the epoxidation of **11** with DMDO cleanly produces **14** [29]. Trials with P450 2E1, P450 3A4 and pHLM all produced similar results. Incubations of **11** with 2E1 and 3A4 yielded a lower proportion of **21** relative to **14**. The reaction of **11** with CAN produces not **14** but the oxidized dimeric molecule **21**, as noted above. Control experiments without the P450 isoforms did not yield **14** or **20**. Incubations with 2E1, 3A4, and pHLM each yield very similar results with 4,5-benzoxepin. They suggest that, for the cytochrome P450s tested, each epoxidize the oxepin structure as the major pathway. It may well be that consecutive single-electron oxidations constitute a relatively minor pathway.

### 2.3. Incubations of 2,7-Dimethyloxepin with P450 Isoforms 

Incubations of 2,7-dimethyloxepin (**8**) with the same suite of enzymes as **11** under the same reaction conditions resulted in inconsistent observations of *E*,*Z*-**10** in very trace amounts. Some of the trials provided dimethylphenols in appreciable yields and dimethyldihydroxybenzenes in low-to-moderate yields. Control experiments suggest that these products are the result of enzymatic reactions, not just thermal rearrangements. There has been no evidence of oxidation at the methyl groups of **8** to form carboxylic acid derivatives. The paper by Golding et al. [30] did not report P450 incubations. As noted earlier, the benzene oxide valence isomer of 2,7-dimethyloxepin is only a few kcal/mol higher in energy than the benzene oxide isomer. Benzene oxides exhibit very short half-lives in protic solvents consistent with the observation of dimethylphenol(s) and even dimethylcatechol(s). In contrast, the epoxide valence isomer (**19**) of 4,5-benzoxepin is energetically inaccessible, shutting down some pathways of competing reactions, thus permitting epoxidation of **11** and its rearrangement to **14**. The GC/MS data for the incubations of **8** are included in the Appendix A.

## 3. Materials and Methods 

Starting materials, reagents, and solvents were purchased from Aldrich, Alfa Aesar and Fisher Scientific and used without further purification unless otherwise indicated. Reaction products were purified by flash column chromatography using Silicycle Inc. silica gel (60 Å, 230–400 mesh). Thin-layer chromatography was performed on Agela Technologies TLC Silica plates (Agela Technologies, Tianjin, China) (silica gel 60 GF254) and visualization was accomplished with UV light (UVP, INC, San Gabriel, CA, USA). 2,7-dimethyloxepin (**8**) and 4,5-benzoxepin (**11**) were synthesized and purified according to the literature [44,45] using standard air-free techniques. Stock solutions of the oxepins 8 and 11 were prepared in acetonitrile and stored at −80°C for use in enzymatic studies. All intermediates and final products were characterized by NMR. ^1^H NMR and ^13^C NMR spectra were acquired with a Varian UnityINOVA 500 NMR (Varian, Palo Alto, CA, USA) or Varian Mercury 400 BB NMR (Varian, Palo Alto, CA, USA). Chemical shifts are reported in parts per million (ppm) relative to tetramethylsilane (TMS) unless otherwise noted and coupling constants (J values) are in Hertz (Hz). Infrared spectra were recorded on a Thermo Fischer Scientific Nicolet iS10 FTIR spectrometer (ThermoFischer Scientific, Waltham, MA, USA). UV–vis data were acquired on a Varian Cary 5E UV–vis-NIR spectrophotometer. NADPH and magnesium dichloride were purchased from Sigma Aldrich (St Louis, MO, USA). Potassium phosphate buffer (0.5 M), pooled human liver microsomes, and cytochrome P450 isoforms 1A2, 3A4, and 2E1 were purchased from Corning Life Sciences in Woburn, MA, USA. MilliQ water was used in all incubations, which were carried out in a 37 °C water bath. All organic extracts of incubation reactions in dichloromethane were characterized using a Shimazdu GCMS-QP2010 (Shimazdu, Kyoto, Japan) equipped with a 30.0 m SHRX1-5MS column (thickness: 0.25 um, diameter: 0.25 um) and 70 eV electron impact detector. The injector temperature (splitless injection) was held at 250 °C and the detector at 260 °C. The column temperature increased from 45 °C to 250 °C at 15 °C/min and was held at 250 °C for 0 min, after a solvent delay of 5.5 min. The flow rate was 1 mL/min and the total run time was 14.67 min. Standards of the reactants and expected metabolites (see Appendix A) were prepared in dichloromethane, and retention times and mass spectra were recorded for comparison.

## 4. Conclusions

The incubation (P450 isoforms and pHLM) experiments employing 4,5-benzoxepin (**11**) clearly indicate epoxidation to form the reactive intermediate 2,3-epoxyoxepin as the major pathway. A lesser metabolic product is the dimeric molecule **20**, a reaction product of the oxidation of 11 by CAN. There was no detectable quantity of **21**, the other CAN product. These observations suggest that cytochrome P450 epoxidizes the oxepin as its major pathway but also catalyzes consecutive one-electron oxidations. Incubation of 2,7-dimethyloxepin with P450 isoforms and pHLM did not produce the anticipated ring-opened diketone. Instead, dimethylphenols and dimethyldihydroxybenzenes were found. Although these results are essentially consistent with the results of Stok et al. [44], who were the first to identify an oxepin as a minor product of incubation of t-butylbenzene with cytochrome P450, it must be noted that this was bacterial P450 rather than mammalian. In contrast to 4,5-benzoxepin, where the naphthalene oxide isomer is energetically inaccessible, the 1,2-dimethylbenzene oxide isomer is only a few kcal/mol higher in energy than 2,7-dimethyloxepin. While it is not self-evident how to differentiate the formation of muconaldehyde from benzene oxide versus oxepin, the work of Golding and colleagues (e.g., Scheme 3) on these tautomers and methyl- and dimethyl-substituted derivatives provides subtle stereochemical probes into these questions. The present work, relying on a “non-natural” metabolite of naphthalene suggests that, for this system, at least, the epoxidation of an oxepin to form a reactive 2,3-epoxyoxepin intermediate is an important, if not dominant, metabolic pathway.

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
