# Peer review of "Cytochrome P450 Can Epoxidize an Oxepin to a Reactive 2,3-Epoxyoxepin Intermediate: Potential Insights into Metabolic Ring-Opening of Benzene"

_molecules, 2020, doi:10.3390/molecules25194542_

Round 1

Reviewer 1 Report

 Weaver-Guevara and Greenberg present the reaction of metabolic ring opening of benzene. The results are useful for the researcher, and the introduction is very clear and complete. Publication after the revision is suggested.

  1. The graphic abstract is full of old researches and should be modified to suit this study.
  2. In Page 2, Scheme 1, if 2,3-epoxyoxopin intermediates cannot be isolated, it should be framed in a bracket.
  3. In ESI, the compound codes do not match the true compounds.
    For example,
    1,4-dihydronaphthalene (23) --- (28)
    4-dihydro-2,3-epoxynaphthalene (24) ---(29)
    1-bromo-1,2-dihydro--2,3-epoxynaphthalene (25) --- (30)

  4. In Page 4, line 127, “Under The”

Author Response

Point 1: The graphic abstract is full of old researches and should be modified to suit this study.

Response 1: The graphical abstract has been revised.

Point 2: In Page 2, Scheme 1, if 2,3-epoxyoxopin intermediates cannot be isolated, it should be framed in a bracket.

Response 2: Scheme 1 includes brackets on the 2,3-epoxyoxepin intermediate which is not observed or isolated, but is a postulated intermediate. 

Point 3: In ESI, the compound codes do not match the true compounds. 
For example, 
1,4-dihydronaphthalene (23) --- (28)
4-dihydro-2,3-epoxynaphthalene (24) ---(29)
1-bromo-1,2-dihydro--2,3-epoxynaphthalene (25) --- (30)

Response 3: In the Supplementary Information, compound codes have been altered for the synthetic sequence of 4,5-benzoxepin to more accurately reflect their “true” compounds.  For example, naphthalene is now labeled 15 (consistent with the manuscript), and the subsequent intermediates to 11 are labeled 34, 35, and 36 respectively as they are not mentioned in the full manuscript.  The compound numbers also match those used in the experimental section and the structure labels on NMR spectra. 

Point 4: In Page 4, line 127, “Under The”

Response 4: Typo has been addressed. 

Reviewer 2 Report

In this manuscript, Greenberg, Weaver-Guevara and co-workers report on the epoxidation of oxepin to 2,3-epoxyoxepin by cytP450. Three cytochrome P450 isoforms as well as pHLM were applied to demonstrate the conversion of 4,5-benzoxepin to 1H-2-benzopyran-1-carboxaldehyde as the major product via 2,3-epoxyoxepin intermediate. Dimethylphenols and dimethyldihydroxybenzenes were formed from 2,7-dimethyloxepin. The study is important to shed light on a potential metabolic pathway of benzene via muconaldehyde. I particularly enjoyed the background on metabolism on benzene and the argument on stability of oxepin isomers. The manuscript is suitable for publication in its present form. The authors are only asked to add compound names below structures in the abstract during the proofing stage.

Author Response

Point 1: In this manuscript, Greenberg, Weaver-Guevara and co-workers report on the epoxidation of oxepin to 2,3-epoxyoxepin by cytP450. Three cytochrome P450 isoforms as well as pHLM were applied to demonstrate the conversion of 4,5-benzoxepin to 1H-2-benzopyran-1-carboxaldehyde as the major product via 2,3-epoxyoxepin intermediate. Dimethylphenols and dimethyldihydroxybenzenes were formed from 2,7-dimethyloxepin. The study is important to shed light on a potential metabolic pathway of benzene via muconaldehyde. I particularly enjoyed the background on metabolism on benzene and the argument on stability of oxepin isomers. The manuscript is suitable for publication in its present form. The authors are only asked to add compound names below structures in the abstract during the proofing stage.

Response 1: No changes needed.